# Homemade Kefir Consumption Improves Skin Condition—A Study Conducted in Healthy and Atopic Volunteers

**DOI:** 10.3390/foods10112794

**Published:** 2021-11-13

**Authors:** Emília Alves, João Gregório, André Rolim Baby, Patrícia Rijo, Luis M. Rodrigues, Catarina Rosado

**Affiliations:** 1CBIOS—Universidade Lusófona’s Research Center for Biosciences & Health Technologies, Campo Grande 376, 1749-024 Lisboa, Portugal; emilia.alves@ulusofona.pt (E.A.); joao.gregorio@ulusofona.pt (J.G.); patricia.rijo@ulusofona.pt (P.R.); 2Department of Biomedical Sciences, Faculty of Pharmacy, University of Alcalá, Carretera Madrid-Barcelona, Km 33.100, 28805 Alcalá de Henares, Spain; 3Department of Pharmacy, Faculty of Pharmaceutical Pharmacy, University of São Paulo, São Paulo 05508-000, Brazil; andrerb@usp.br; 4Instituto de Investigação do Medicamento (iMed.ULisboa), Faculdade de Farmácia, Universidade de Lisboa, 1649-003 Lisboa, Portugal

**Keywords:** kefir, cutaneous health, atopic dermatitis, transepidermal water loss (TEWL), hydration, skin barrier, scoring of atopic dermatitis (SCORAD)

## Abstract

Diet has a fundamental role in the homeostasis of bodily functions, including the skin, which, as an essential protective barrier, plays a crucial role in this balance. The skin and intestine appear to share a series of indirect metabolic pathways, in a dual relationship known as the “gut-skin axis”. Hence, the gut-skin axis might be receptive to modulation via dietary modification, where probiotics can be included, thus representing a potential therapeutic target in inflammatory skin diseases, such as atopic dermatitis (AD), in order to control and/or ameliorate symptoms. Kefir is one of the most ancient fermented foods, with probiotic characteristics that have been associated with a wide variety of health-promoting benefits, and it presents a microbiological diversity that makes its application as a probiotic in the gut-skin relationship of the utmost interest. However, the impact of a diet containing kefir on skin health has yet to be reported in scientific literature. This study aimed to assess the impact of the intake of homemade kefir in the skin of healthy and atopic volunteers. The intervention resulted in a boost on barrier function in both skin types verified only in the respective kefir intake groups. An improvement in the degree of severity of AD was also confirmed for the kefir intake group. Atopic individuals may benefit from kefir intake, especially in regard to their skin hydration. Finally, the effects observed on skin barrier function in this study probably culminate from the effects of all the ingredients in kefir, including the complex microbiota, its metabolites and macro- and micronutrients resulting from the fermentation. This work opens the way for more advanced research on the impact of the probiotic kefir on cutaneous health, further clarifying its mechanism of action namely via gut-skin axis.

## 1. Introduction

Diet has a fundamental role in the homeostasis of bodily functions, including the functions of the skin, which, as an essential protective barrier, plays a crucial role in this balance [1,2,3]. The skin and intestine appear to share a series of indirect metabolic pathways in a dual relationship known as the “gut-skin axis” [4,5,6]. On the one hand, the impairment of the intestinal microbiota is linked to the development of allergic diseases, and the intestinal microbiota and/or dietary metabolites can be detected in the skin. On the other hand, skin health has been linked to the integrity of the intestinal barrier and/or suppression of pro-inflammatory mediators, e.g., via vitamin D [4,7,8,9].

In recent years, growing research in these areas of interest within human nutrition has led to the expansion of probiotics as health promoters [10,11]. Probiotics are live microorganisms that, by definition, must confer a health benefit on the host. Probiotics may act via numerous mechanisms, including the restoration of intestinal microbial balance, prevention of pathogen invasion by competitive binding to epithelial cells, suppression of pathogen growth by bacteriocin secretion, and restoration of impaired intestinal barrier function [10,12]. Given that the gastrointestinal mucosa and gut-associated lymphoid tissue harbor more than 70% of the body’s immune cells, this may explain the growth in research data linking these organs to multiple disease mechanisms. This seems to be the case in atopic dermatitis (AD), one of the most prevalent inflammatory skin diseases [6,7,13].

AD has been associated with an exacerbated skin response to environmental agents, characterized by relevant symptoms including pruritic lesions with typical morphology, pain, and sleep disturbances [1,14]. The onset of AD points towards a complex interaction between skin barrier dysfunction, immune dysregulation, environmental risk factors, and (intestinal and skin) dysbiosis [1,5,6,15]. Intestinal dysbiosis seems to increase epithelial permeability via pro-inflammatory cytokines, promote immune dysregulation, and intensify the chronic systemic inflammation in AD [4,5,6,7,9,16]. Conversely, this also suggests that the gut-skin axis would be receptive to modulation via dietary modification, wherein probiotics can be included, thus representing a potential therapeutic target in AD to control and/or ameliorate AD symptoms [11,17,18,19].

Kefir is one of the most ancestral fermented foods with probiotic characteristics [20,21,22,23]. Traditionally prepared by the fermentation of milk with kefir grains and most popular in northeastern Europe and Asia, it consists of a symbiotic mixture of lactic acid bacteria (LAB) and yeasts that, in addition to acting synergistically, also produce several bioactive compounds [24,25,26,27]. A wide variety of health-promoting benefits have been associated with its use [28,29,30,31,32,33] and have expanded its popularity beyond its traditional borders within northeaster Europe and Asia. Anti-inflammatory effects [34], antimicrobial activity [35], strengthening of the immune system [36], antioxidant activity [37], and the inhibition of pathogenic microorganisms [24,38] have been reported as a result of kefir consumption. In addition, the topical application of a gel made from a non-microbial fraction of kefir showed an improvement in the wound healing capacity [39]. These properties have been attributed both to the presence of a complex microbiota, with high resistance to passage through the gastrointestinal tract and high adhesion capacity to the intestinal mucus, and to the action of metabolites released during fermentation, namely organic acids and short chain fatty acids (SCFA) [24,37,40,41,42]. Nevertheless, studies demonstrating its therapeutic interest in specific conditions are very limited, and its potential skin benefits and applicability in the management of AD have yet to be explored.

This study aimed to assess the impact of the regular consumption of kefir prepared in homemade conditions in the skin of healthy and atopic volunteers.

## 2. Materials and Methods

### 2.1. Study Design

A controlled intervention study, coded DermapBio, was conducted according to the principles of the Helsinki Declaration and after informed consent. The study protocol was approved by the ethics committee of the School of Sciences and Health Technologies at Lusofona University (N°1/2018, 15 May 2018). Study subjects were recruited by convenience sampling between October 2019 and December 2020. Subjects were asked to answer a questionnaire that examined sociodemographic and lifestyle conditions, as well as specific inclusion/exclusion criteria. These criteria are summarized in Table 1.

Subjects were assigned to the different groups according to the inclusion criteria. The atopic group (*n* = 19) included 1 male and 18 females, aged between 19 and 56 years (mean age 31.7 ± 11.9 years), wherein 47% were under 30 years old. Within this group, all subjects along with presented with AD; 14 (74%) of the subjects within this group also reported rhinitis, and 6 (32%) reported asthma diagnosis. All other subjects who fulfilled the eligibility criteria, excluding the atopic criteria, and were free of skin diseases, including AD, psoriasis, and other systemic diseases that may impact skin condition, were assigned to the healthy group. These subjects (*n* = 33) included 6 males (18%) and 27 females (82%), aged between 20 and 60 years (mean age 27.0 ± 10.1 years), wherein 61% were under 30 years old. Within each group, volunteers were assigned to either the kefir intake or the control (without intake) group, according to their preference.

This research aimed to compare, for each skin type evaluated, the effect of kefir ingestion between the intervention groups and respective controls, thus using a parallel group design. However, this design is unable to distinguish between changes induced by food ingestion and those induced by differences between individuals at baseline [43]. Therefore, in order to minimize baseline individual variability, especially in studies involving dietary interventions, a crossover design is recommended as individuals are used as their own controls (paired comparisons) [43,44]. Hence, a crossover design was then sequentially applied to each study group, since, for each individual, a comparison was made between the parameters measured before and after the intervention (Figure 1).

All subjects were instructed to proceed as follows during the study period: avoid over-exercising and major lifestyle changes; not consume dietary supplements or fermented foods; not change their usual dietary intake of food fiber or food containing oligosaccharides; refrain from using laxatives; refrain from changing type and frequency of regularly used skin-care agents; avoid travelling abroad.

Physiological conditions, skin phototype, and anthropometric measurements (weight, height, and waist circumference) were obtained from all the participants. The skin phototype was assessed (by a single researcher) using the Fitzpatrick phototype classification [45]. Height was self-reported, weight was measured using a digital weight scale Tanita^®^ BC601 (Tanita Europe BV, Amsterdam, the Netherlands), and waist circumference (WC) was measured using a Kern^®^ MSW circumference tape measure (KERN & SOHN GmbH, Balingen, Germany). Body Mass Index (BMI) was calculated as weight (kg)/(height (m))^2^ [46].

The control groups, A0 and H0, did not consume kefir. The intervention in groups AK (atopic skin with kefir intake) and HK (healthy skin with kefir intake) consisted of the daily consumption of kefir for eight weeks. This period has been adopted in similar trials [47,48,49,50,51,52], and is supported by the fact that approximately two weeks are required for the development of a consistent probiotic gut colonisation, i.e., stable detection in faecal content, and approximately one month to observe a significant change in cytokines at the gut level. Thus, the period of eight weeks, being sufficient to impact the bowel, would also be long enough for a putative effect on the skin.

The primary endpoints in this study were a decrease in transepidermal water loss (TEWL) and an increase in *stratum corneum* (SC) hydration for all subjects, and a decrease in the SCORAD Index for atopic subjects.

### 2.2. Assessment of Dietary Intake

Dietary intake was assessed at baseline, for all subjects, through a three-day dietary record (two weekdays and one weekend day) [53,54]. Detailed instructions for record-keeping were provided in writing to all subjects.

### 2.3. Kefir Intervention

Kefir grains CIDCA AGK1 were obtained from the *Centro de Investigación y Desarrollo en Criotecnología de Alimentos* (CIDCA), La Plata, Argentina. Microbiological characterization, preservation, and storage of these grains have been described elsewhere [55,56,57]. Kefir was produced by fermentation of a commercial ultra-high temperature pasteurized (UHT) semi-skimmed cow milk of Portuguese provenance (Nova Açores^®^, S. Miguel, Portugal), with CIDCA AGK1 kefir grains using a grain inoculum of 10% (*w*/*v*), for 24 h, at a temperature of 20 ± 1 °C. The fermentation conditions were designed to be representative of Portuguese household conditions, as described elsewhere [27]. In order to assure the daily intake of kefir for eight consecutive weeks, each subject of both intake groups (AK or HK) visited the research center three times a week (Monday, Wednesday and Friday). During the visit, the subjects drank 100 mL of kefir and were given white plastic sterile containers with the kefir doses of the following days. The subjects were instructed to store these kefir samples in their household refrigerator to maintain their characteristics [27]. It was determined that 100 mL of the prepared kefir had a nutritional composition of 1.28 ± 0.04 g of fat, 3.15 ± 0.19 g of protein and 4.91 ± 0.19 g of carbohydrates and 0.6 g of lactic acid. Microbiologically, it provided 7 × 10^9^ colony-forming units (CFU) of LAB and 2 × 10^8^ CFU of yeast [27], which is consistent with the literature for the daily ingestion of probiotic bacteria capable of surviving passage through the gastrointestinal tract and thus reaching the necessary sites to exercise their positive physiological functions, both intestinal and immunological [58,59].

### 2.4. Skin Measurements

The skin condition was quantitatively evaluated by non-invasive bioengineering equipment, including those assessing TEWL, SC hydration, and erythema, which is a sign of exacerbation in AD [60,61,62]. TEWL, a measure of the rate of water lost through the skin, reflects barrier dysfunction directly, thus being a parameter of interest to evaluate skin barrier function in both healthy and diseased skin [60,63]. It was measured using a Tewameter^®^ TM300 (Courage + Khazaka Electronic GmbH, Köln, Germany) in accordance with the published guidelines [64], and measurements were expressed as g/m^2^/h. Skin hydration is indicative of the water content of the SC, which is also a parameter of interest in both healthy and atopic skin [61,62]. It was measured using a Corneometer^®^ CM825 (Courage + Khazaka Electronic GmbH, Köln, Germany) and was assessed as skin conductance given by the reactive capacitance of skin, using the stratum corneum as a dielectric membrane [65]. Measurements were expressed in arbitrary units (AU).

All participants were advised to refrain from using moisturizers or other cosmetic products in the tested areas 48 h before the measurements. Measurement areas were assigned in the ventral forearm (10 cm below the inner elbow crease), leg (outer side, 10 cm below the knee), and forehead (mid area). Measurements were taken in all subjects before and after the eight weeks of intervention, t0 and t8, respectively, and were performed by the same researcher using identical standards. Measurements were performed under controlled temperature (21 ± 1 °C) and humidity conditions (relative humidity, 50 ± 10%) after a period of acclimatization of 20 min.

As shown in a previous study, the use of a stress test to assess the skin barrier function after a probiotic intervention represents a novel approach in this field [63]. Therefore, a sodium lauryl sulfate (SLS)-induced skin lesion model was applied at baseline (t0) and after the intervention period (t8). This test, consisting of the application of a 1% solution of SLS under occlusion for 24 h, was conducted in the forearm and only on volunteers with healthy skin, as the application of SLS would be detrimental to the volunteers in the atopic group, potentially causing excessive discomfort. The extent of the impact of SLS was assessed by evaluation of TEWL combined with measurement of erythema as described elsewhere [63] using a Chroma Meter^®^ CR300 (Konica Minolta, Tokyo, Japan) and expressed as a* in the L*a*b* system color [66].

### 2.5. SCORAD Index Assessment

The standard scoring system of Atopic Dermatitis—SCORAD Index, developed by the European Task Force Group on Atopic Dermatitis (ETFAD), considered the best validated scoring system to assess AD clinical severity, was applied in this study [67,68]. This severity classification system contemplates two distinct scores: the objective SCORAD score (intensity and extent of the lesions), which ranges from 0 to 83; and the subjective SCORAD score (pruritus and sleep loss), which extends the SCORAD total score to a maximum of 103. The objective SCORAD score is divided into part A, consisting of the interpretation of the extent of the disorder, which represents the affected body sites, and part B representing the intensity of the lesions. The subjective SCORAD score is given by part C, consisting of symptoms such as itching and sleep loss during the three days prior, and is scored by the patients. SCORAD Index is determined by [68]:$$SCORAD\;Index=\frac{A}{5}+\frac{7B}{2}+C$$

The SCORAD Index was assessed at t0 and t8 (only) in the atopic group by the same researcher using identical criteria. Considering the ETFAD recommendation, the AD severity was classified as mild for SCORAD Index < 25, as moderate for SCORAD Index between 25–50, and as severe for SCORAD Index > 50 [67].

### 2.6. Statistical Analysis

Results were expressed as mean ± standard deviation (SD), as relative frequencies, or as median and first and third quartiles. Since the data were not normally distributed (normality assessed by the Shapiro–Wilk test), non-parametric tests were chosen to test different hypotheses. For continuous variables, differences within individuals were identified by Wilcoxon signed rank test and differences between kefir intake and control groups by Mann-Whitney U test. The Chi-square test was used to test associations between categorical variables. Pearson’s correlation was used to evaluate possible relations between skin barrier function parameters and the severity of AD. Linear regressions were used to evaluate the association between kefir intake and skin improvements and their potential confounding factors. All analyses were performed using the SPSS statistical package version 25 (SPSS Inc., Chicago, IL, USA) with a significance level of 0.05.

## 3. Results

### 3.1. Study Groups Characteristics

Before the beginning of the intervention, socio-demographic characteristics were assessed (Appendix A). Data concerning daily food intake were also collected (Appendix A). The physiological characteristics are presented in Table 2.

Despite the different sample sizes of the groups, subjects who were given kefir, either healthy or atopic, showed no differences in physiological characteristics, regarding the respective control groups, at baseline, as shown in Table 2. In addition, no differences were found for lifestyle indicators, such as cigarettes and alcohol consumption, nor dairy intake (Appendix A). All groups presented a mean BMI below 24.9 kg/m^2^, representative of normal weight, and a mean WC below 80 cm, considered within the normal range for both men and women, thus indicating low risk of metabolic diseases [69]. Regarding the dietary intake, no differences were observed for energy, macronutrients, and water, between subjects who drank kefir, either healthy or atopic, and their respective controls (Appendix A). Although all macronutrients (assessed as a percentage of the energy intake) were within the recommended range, fiber intake was found to be lower than the recommendation [70]. These data indicate identical baseline characteristics and conditions for the intervention among kefir intake and control for both healthy and atopic groups.

### 3.2. Skin Measurements

Skin condition was assessed by measuring TEWL, SC hydration, and erythema, at t0 and at t8.

An analysis of the variation of skin parameters after eight weeks was conducted, comparing kefir intake and control groups in both healthy and atopic volunteers (Table 3). In order to minimize the impact of interindividual variability, for this comparison, variation on skin parameters was computed as a deviation from baseline, calculated as:$$Deviation\;\left(variable\right)\;=\;\left[\left(variable\;at\;t8\right)\;-\;\left(variable\;at\;t0\right)\right]/\left(variable\;at\;t0\right)$$

Deviation variables must be interpreted as follows: for TEWL and erythema, negative values represent an improvement in skin condition after the intervention, while for hydration, an improvement is only observed when the deviation value is positive.

As shown in Table 3, on the healthy skin volunteers, forearm and forehead TEWL decreased in the HK group compared to H0 group (*p* = 0.018 and *p* = 0.036, respectively). Moreover, the kefir-supplemented group showed increased forearm and forehead hydration, compared to the control (*p* = 0.034 and *p* = 0.012, respectively). No differences were observed between HK and H0 for erythema (*p* = 0.685). These results are supported by those obtained in individual paired comparisons that showed that after eight weeks of kefir ingestion, on healthy subjects, forearm and forehead TEWL and erythema decreased significantly compared to t0 (*p* = 0.016, *p* = 0.019, and *p* = 0.023, respectively) (Appendix A), thus confirming the effective improvement in skin conditions.

Furthermore, results from application of the SLS induction lesion model on healthy skin, performed at t0 and t8, are shown in Table 4.

The results from Table 4 showed a significant decrease in TEWL on the HK group compared to control (*p* < 0.001), thus corroborating the above-mentioned results for forearm TEWL, shown in Table 3.

For atopic skin subjects, variations on skin parameters were noted in the AK group after eight weeks (Table 3). TEWL decreased in the forearm (the more significant change), forehead, and leg, compared to the A0 group (*p* < 0.001, for all cases). Regarding hydration, the AK group showed an increase in forearm and leg compared to control (*p* = 0.001 and *p* = 0.034, respectively). No differences were observed for erythema (*p* = 0.221) between these groups (Table 3). These results were reinforced by those from individual paired comparisons that showed that at t8, the atopic subjects who drank kefir presented a significantly lower TEWL and erythema and a significantly higher hydration compared to t0 in all anatomical study areas (*p* < 0.05, for all parameters), while in the control group, no differences were observed (Appendix A), thus confirming the effective change in skin conditions, despite individual baseline conditions.

### 3.3. SCORAD Index Assessment

The SCORAD Index was evaluated at t0 and t8 for all subjects from the atopic group. Variation on the SCORAD Index was assessed as a deviation and was computed using the previously explained approach; thus, negative values represent an improvement in AD symptoms after the intervention. The results are shown in Table 5.

As shown in Table 5, after eight weeks of kefir intake, the AK group showed a significant decrease in the SCORAD Index, compared to control (*p* < 0.001). These results were corroborated by those of paired individual comparisons in which at t8 atopic individuals who drank kefir had a significantly lower SCORAD index compared to t0 (*p* < 0.05), whereas for control individuals, no differences were observed (see Appendix A), thus confirming the effective change in skin conditions despite individual baseline conditions.

It is noteworthy that, at the beginning of the study, the AK group presented a median SCORAD Index value of 61.9 (41.2, 72.2), a range classified as severe AD, while after the intervention with kefir, the median SCORAD Index was 16.2 (12.1, 32.9), a range classified as mild AD, according to the ETFAD recommendation [67]. As for group A0, the median SCORAD Index value was 33.4 (23.1, 52.1) at t0 and 33.9 (26.8, 69.4) at t8, thus classifying the severity of AD as moderate both at baseline and after the intervention period.

Furthermore, in atopic volunteers, a possible relationship between cutaneous parameters and the AD severity was also assessed using Pearson’s correlation. We observed a significant correlation between the improvement of AD severity given by the deviation of the SCORAD Index and skin barrier improvement given by the deviation of TEWL on the forearm, leg, and forehead (r = 0.630, *p* = 0.004; r = 0.481, *p* = 0.037; r = 0.680, *p* = 0.001, respectively), and also on the forearm hydration (r = −0.839, *p* < 0.001). Although erythema, a well-known sign of skin inflammation, is present in both acute and chronic stages of AD [60], our results were not able to detect a relation between erythema and the improvement of AD severity (r = 0.286, *p* = 0.236).

### 3.4. Adjusted Models for Skin Parameters

To assess the effect of different independent variables on the outcomes of skin parameters, multiple linear regression models were performed. All socio-demographic variables, food intake variables, kefir intake, and skin status were considered as possible predictors for the influence in deviation of skin parameters. After testing the assumptions for linear regression and collinearity diagnostics, independent variables were excluded from the models if the variance inflation factor (VIF) was superior to 10. Following this step, backward stepwise linear regressions were performed for each outcome variable to identify which variables better explained the outcome variable. Although the climatic conditions (temperature and humidity) were evaluated at t0 and t8, they were not used in the regression models as they did not affect the effect of kefir intake (*p* = 0.329, *p* = 0.464, *p* = 0.352 and *p* = 0.363, respectively), thus not being considered relevant.

The most common variables in the models and so considered as possible predictor variables on skin parameters identified by this method were: kefir status, defined as with or without kefir intake; skin status, defined as belonging to the healthy or the atopic group; gender, defined as male or female; and water intake in liters. New linear regressions with the Enter method were then run, which are presented in Table 6.

The results from Table 6 show that drinking kefir for eight weeks is associated with a significant improvement in TEWL and in SC hydration, in all study areas (Model 1). In the adjusted models for skin status (Model 2) and skin status, gender, and water intake (Model 3), the effect of kefir intake remained significant, continuing to show an improvement in TEWL and hydration, with the best results obtained for the forearm. The SCORAD Index clearly improved with kefir intake.

## 4. Discussion

New insights in the field of nutrition increasingly support the evidence of a close relationship between diet and health, namely skin health [2,71]. Diet is a major regulator of the intestinal microbiota, and short-term changes in the diet have the ability to rapidly alter gut bacteria [44,72]. The use of probiotics presents itself as one of the most common interventions to beneficially regulate the gut microbiota [2,10,71]. Probiotics are beginning to be recognized as being able to beneficially impact skin health by modifying its microbiota, preventing pathogen invasion and contributing to the restoration of impaired barrier function [13,73,74,75].

The consumption of kefir has been reported to positively impact the gut microbiota and overall condition of the digestive system [33,76,77,78]. Additionally, an in vitro study suggests that kefir’s passage through the human gastrointestinal tract, and its consequent digestion, can improve its nutritional profile and bioactivity [79]. However, to date, most studies aiming to establish the benefits to human health of kefir consumption have been based in animal models, or in cell culture systems wherein the digestion of kefir does not occur, thus providing limited information [28,79]. Of note, none of the in vivo human studies found in the literature observed the skin impact of a diet containing traditionally homemade kefir as the probiotic, neither in healthy nor atopic subjects.

In this study, kefir intake for eight weeks caused an improvement in the skin condition of healthy subjects, quantitatively demonstrated by a significant decrease in TEWL and increase in hydration on the forearm and forehead, compared to the control. Similar results were found in other in vivo studies evaluating the effect of ingested specific probiotic strains (*Lactobacillus* and *Bifidobacterium* species) in human adults with healthy skin [48,49,80,81,82]. Kano et al. and Mori et al. evaluated the effect of ingesting fermented milk containing one strain of *Bifidobacterium* species for eight weeks. They both found a significant improvement on SC hydration in the probiotic ingestion group, and attributed their results to an improvement in intestinal conditions, as the levels of toxic metabolites excreted by intestinal bacteria such as phenol decreased [81,82]. In the study by Gueniche et al., a significant decrease in TEWL was observed after eight weeks of probiotic intervention [48]. Moreover, Ogawa et al. found a significant decrease in TEWL and an increase in SC hydration after twelve weeks of probiotic intake [49], and Lee et al. observed identical results after twelve weeks of probiotic intake [80]. However, not all skin studies using probiotics have been able to demonstrate this type of outcome. Saito et al. tested the ingestion of one probiotic strain (*Lactobacillus* species) by healthy volunteers and found a decrease in TEWL at the arm, but not the face, and was not able to detect changes in skin hydration [47].

An innovative note in our approach is the use of the SLS irritation induction model to further demonstrate the beneficial impact of kefir consumption in barrier function in healthy skin. These results are supported by previous research by the authors [63]. Other studies using similar approaches but conducted in animal models exposed to irritants (ex vivo and in vivo) observed a decrease in TEWL after ingesting probiotics [83,84].

Atopic dermatitis (AD), the most common form of eczema, is a chronic inflammatory skin disease characterized by symptoms such as pruritic lesions, pain, and sleep disturbances [1,15]. AD onset points towards a complex interaction between skin barrier dysfunction, immune dysregulation, environmental risk factors, and dysbiosis of the intestinal and skin microbiota, which correlates with the clinical severity of AD [1,14,15,61]. Through the gut-skin axis, intestinal dysbiosis has been shown to negatively impact skin function either through an increase of epithelial permeability, via pro-inflammatory cytokines, thus promoting immune dysregulation and contributing to the chronic systemic inflammation in AD, as by perpetuating pruritus via secretion of neuroendocrine itch mediators, leading to a chronic itch–scratch cycle, thus further disrupting the skin barrier [1,5,6,9].

In AD, the presence of an impaired epidermal skin barrier is demonstrated by both a defective inside–outside barrier (increased TEWL) as well as a defective outside–inside barrier (increased penetration of environmental substances triggering immunological mechanisms), along with decreased hydration of the SC [1,14,15]. A lower content in SC ceramides, unsaturated fatty acids, and structural proteins such as filaggrin (involved in SC barrier formation and hydration) underlies the cutaneous barrier dysfunction in AD [1,74,85]. Traditional therapy used in AD is based on topical treatments, often corticosteroids, thus being focused on treating symptoms rather than the underlying causes, therefore mainly resulting in a short-term repair of the defective barrier [1,85,86].

Although to date several studies have explored the potential efficacy of probiotics in the prevention and treatment of AD, the results are not consistent, thus contributing to the lack of evidence for the use of probiotics in skin health [11,73,75,85,86,87]. The variation in types of strains used, both in diversity and in doses, different types of formulation (supplement or food), duration of the ingestion period, as well as the type of parameters used to assess skin conditions can somehow justify this lack of consistency in the results obtained [86].

In our study, a significant decrease in TEWL and increase in hydration was observed in subjects with AD who drank kefir for eight weeks in all the anatomical areas of study, which was not observed in the controls. Furthermore, our data also showed a significant decrease in the SCORAD index in the kefir ingestion group compared to controls, with the level of AD severity changing from severe to mild, which reflects a notable clinical improvement. These results are in agreement with similar in vivo studies conducted on other probiotics [50,51,88,89]. In a randomized cross-over study using a combination of the probiotics (*Lactobacillus* and *Bifidobacterium* species) delivered as food (yogurt) for eight weeks, Roessler et al. observed a non-significant decrease in SCORAD in the atopic group only [50]. Similarly, Yoshida et al. supplemented adults with AD for eight weeks using a capsule formulation with one probiotic strain (*Bifidobacterium* species) and found a significant decrease in SCORAD only in the probiotic intake group, which was attributed to changes in intestinal microflora [51]. In another study, Iemoli et al. found an improvement in SCORAD in adults with AD after a twelve-week intake of a freeze-dried powder mixture of two probiotics (*Lactobacillus* and *Bifidobacterium* species), and justified this by an improvement in the immune response, namely, by the increased production of T-helper cell type-2 (Th2) and regulatory T cells, and by the reduction of microbial translocation in the intestine [90]. Drago et al. observed identical results after a 16 week intervention with sachets containing one *Lactobacillus* species, in AD volunteers, and attributed them to a significant decrease of T-helper cell type-1 (Th1) inflammatory cytokines and Th1/Th2 ratio [88]. These studies highlight the microbiota’s ability to impact the lymphoid tissue associated with the intestine, via microbial–mucosal interaction [6,13]. To date, the only meta-analysis performed evaluating the effect of oral probiotics in adults with AD found an overall improvement in the SCORAD index (−8.26, 95% CI: −13.28, −3.25) favoring probiotics [87]. Moreover, a minimum intervention time of eight weeks has also been shown to be adequate to assess the impact of probiotics on AD [87]. Finally, and in contrast to our study, Matsumuto et al. were not able to find any differences in AD severity between probiotic and control groups using one strain of *Bifidobacterium* delivered in the form of capsules to AD patients for eight weeks [52].

Furthermore, in studies that assess the impact of probiotics on AD, typically, only clinical parameters are evaluated, usually severity using the SCORAD Index or equivalents [11,75,86,90]. Our approach, combining clinical and skin barrier function assessment, revealed a strong correlation between the improvement in both the severity of skin lesions and TEWL, thus confirming previous reports on the relationship between TEWL and the clinical status of patients with AD [60,89].

Additionally, no differences in erythema were observed between the study groups in our study, which can be explained by the fact that although erythema is particularly associated with acute skin inflammation, it can also be present in both the acute and chronic stages of the disease; similar results were found in the literature [60]. This may be indicative that erythema measurement is not as sensitive as the measurement of TEWL and so it may not be useful to detect subclinical lesions.

The set of results obtained showed that for both skin types, subjects who drank kefir for eight weeks presented a significant improvement in skin barrier function. Among volunteers who consumed kefir, it is noteworthy that the greatest improvement in both TEWL and hydration was observed in atopic individuals, especially in the forearm. These results also show the relevance of evaluating different anatomical areas in skin studies. Among all anatomical study areas, the forearm showed to be the most sensitive area in obtaining skin variations, for both TEWL and hydration. Such variations observed in the cutaneous parameters in different anatomical regions may be related to differences in thickness and also at the SC level, namely in ceramides and filaggrin; as well as differences in the cutaneous microcirculation [91]. Probiotics may positively impact the skin by enabling the production of bioactive bacterial compounds such as lactic acid, hyaluronic acid, and SCFA [42,74]. We have previously demonstrated that the kefir produced under home use conditions used in this study fulfills the lactic acid requirement for a fermented product [27].

The concept of hormesis can also contribute to justify our results, since it is a biphasic dose/concentration response, characterized by a low-dose stimulation and a high-dose inhibition, based on adaptive responses of biological systems to moderate or self-imposed environmental challenges, whereby the system improves its functionality and/or tolerance to more severe challenges [92]. Calabrese et al. found that while normal and high-risk groups generally exhibit hormonal dose responses to the same inducing agent, high-risk groups tend to respond better to lower doses [93].

Exposure of probiotic LAB to stressors, both during fermentation and in the gastrointestinal tract, affects its survival, as well as its proliferation and gastrointestinal functionality [94]. In the intestine, the probiotic LAB exhibits substantial antioxidant activity, promoting the production of antioxidant enzymes, thereby helping to remove ROS and alleviating oxidative stress [94]. Furthermore, improved survival of the probiotic LAB during fermentation is achieved by co-culture with initial strains. Given that yeasts have greater antioxidant activity than LAB, when used in co-culture, there is an increase in antioxidant activity, growth rate, and protective effect against oxidative damage [95]. Exposure of a probiotic strain to a sublethal level of oxidative stress will induce an adaptive response and improve the strain’s resistance to potentially higher levels of oxidative stress. Probiotic bacteria can exert their antioxidant activity through the scavenging of free radicals, chelation of metal ions, enzymatic regulation, and modulation of the intestinal microbiota [37,94].

It can be highlighted that in AD, chronic skin inflammation is associated with the overproduction of reactive oxygen species (ROS), such as superoxide (O^2−^) and hydrogen peroxide (H_2_O_2_), thus generating an oxidative stress condition [96,97]. It is known that mitochondria play an essential role in both homeostasis and inflammatory conditions of the skin [98]; thus, the mitochondrial dysfunction of the skin, caused by the production of ROS, is a potential contributer to the mechanism of AD initiation [99].

The effects observed on skin barrier function in this study can likely be attributed to the combined effects of all kefir ingredients, including its complex microbiota, its metabolites, and macro- and micronutrients resulting from the fermentation, as eating a food promotes a whole-body effect [38,40,41,100].

However, although the kefir grain microbial composition is very stable [55,100], the concentration of metabolites and inhibitory compounds that interact with each other may differ in every fermentation process, thus in part justifying the different results reported in the literature. Of note, no adverse effects were reported during the kefir intake period in this study. In addition, unlike many previous reports, our work was conducted in vivo in humans, which probably highlights the effect of kefir digestion in putative health benefits.

Despite all the positive outcomes found in our study, some limitations must be acknowledged. First, this was not a double-blind, placebo-controlled study. Furthermore, although the study design was intended to minimize the effect of individual variability and the small number of participants, individual changes can occur over time, influencing the dynamics of the gut-skin axis and thus impacting the results [44]. However, these challenges can be mitigated by introducing a washout period and collecting new baseline samples before starting a second sequential intervention.

Moreover, although we did not identify any relationship between nutrient intake and the measured skin parameters, that influence is expected to exist due to the impact of food in the gut, particularly fiber and water intake [2,44].

Along with the proven utility of the determination of TEWL and SC hydration, evaluation of the skin barrier function should also include assessment of the content of other relevant components of SC, with a focus on the ceramides profile, as well as a determination of the impact of the probiotic intake in the skin microbiota [5,14,74,87].

The ability of probiotics to modulate the gut microbiota and the immune status suggests that systemic immunomodulation occurs following ingestion [4,10]. Although all probiotics must present common properties such as low pathogenicity, resistance to gastric acid and bile salt digestion, and adherence to intestinal mucosa, their clinical effects may be species-dependent [10,11]. Therefore, monitoring changes in the human gut microbiome after ingesting a multi-strain probiotic, such as kefir, can provide a better understanding of the mechanisms underlying its many health benefits. Finally, conditions affecting the kefir production, such as the fermentation conditions or origin of the grains, should be considered for in-depth analyses regarding the impact of kefir on health [27].

## 5. Conclusions

We investigated the effects of ingestion of homemade kefir on the skin condition, as well as on the SCORAD Index of the atopic individuals. Our results showed a significant improvement on all skin outcomes and suggest that atopical individuals may benefit from kefir intake, especially regarding their skin hydration. The nutritional and microbial richness of kefir makes its application highly relevant within many sectors of health care.

To the best of our knowledge, this was the first study to provide information regarding the cutaneous impact of the intake of kefir produced in household representative conditions. Furthermore, the simultaneous improvement seen in all skin parameters observed in this study is considered a new finding.

This work opens the possibility of continuing the research of the impact of the probiotic kefir on cutaneous health and its mechanism of action via the gut-skin axis.

## Figures and Tables

**Figure 1 foods-10-02794-f001:**
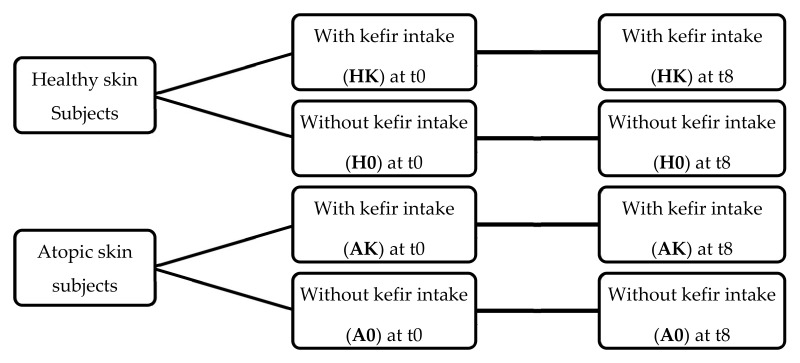
Study design regarding skin conditions and kefir intake during the eight-week intervention period: HK—healthy skin with kefir intake; H0—healthy skin without kefir intake; AK—atopic skin with kefir intake; A0—atopic skin without kefir intake.

**Table 1 foods-10-02794-t001:** General inclusion and exclusion criteria and atopic group specific inclusion criteria.

General Inclusion criteria
Volunteers of both genders aged between 18 and 64 years old
Atopic inclusion criteria
Eczema/atopic dermatitis diagnosis Rhinitis or allergic conjunctivitis diagnosis Asthma diagnosis
General Non-inclusion/Exclusion criteria
Regular consumption of kefir or any probiotic strains (as supplements or pharmaceuticals) in the three months prior to the study or during the study Oncologic disease Women who were pregnant or breastfeeding Gastrointestinal disease diagnostic affecting bowel movement (such as Irritable Bowel Syndrome or Crohn’s Disease) Retinoid treatment in the three months prior to the study or during the study Antibiotic treatment in the 30 days prior to the study or during the study Topical treatment with corticosteroids/anti-inflammatories in the study area in the eight days prior to the study or during the study Chronic illness that involves taking regular (daily) medications such as insulin, oral antidiabetics, anti-inflammatories, or immunosuppressants Skin disease in the study areas Cosmetic treatment of the skin, scrubbing, or depilation at the study areas in the 30 days prior to the study, or during the study period Failure to comply with the guidelines of the study

**Table 2 foods-10-02794-t002:** Physiological characteristics of study participants (relative frequency (%); mean ± SD).

Physiological Characteristics	Healthy Group (*n* = 33)			Atopic Group (*n* = 19)		
	HK	H0	*p*-Value	AK	A0	*p*-Value
Gender			0.208			0.330
Female, n (%)	12 (92.3)	15 (75.0)		9 (100)	9 (90.0)	
Male, n (%)	1 (7.7)	5 (25.0)		0	1 (10.0)	
Age, mean (SD), years	28.9 (13.0)	25.8 (7.71)	0.739 *	30.4 (12.3)	32.9 (12.1)	0.538 *
Skin Phototype			0.388			0.252
Type II, n (%)	6 (46.2)	7 (35.0)		4 (44.4)	2 (20.0)	
Type III, n (%)	5 (38.5)	12 (60.0)		5 (55.6)	8 (80.0)	
Type IV n (%)	2 (15.3)	1 (5.0)		0	0	
BMI, mean (SD), kg/m^2^	22.6 (3.68)	23.3 (4.19)	0.439 *	22.7 (3.40)	22.8 (2.18)	0.540 *
Waist circumference, mean (SD), cm	72.4 (9.19)	77.5 (13.6)	0.328 *	77.2 (8.67)	78.6 (6.02)	0.653 *

SD—standard deviation. BMI—Body Mass Index. HK—healthy skin with kefir intake; H0—healthy skin without kefir intake; AK—atopic skin with kefir intake; A0—atopic skin without kefir intake. Groups compared by Chi-square test, except (*) where Mann-Whitney U test was applied, with *p* < 0.05 for statistical significance.

**Table 3 foods-10-02794-t003:** Comparison of skin parameters variation, between kefir intake and control groups, for both healthy and atopic volunteers, after eight weeks of kefir ingestion (median (Q1, Q3)).

Deviation of Skin Parameters	Healthy Group (*n* = 33)			Atopic Group (*n* = 19)		
	HK	H0	*p*-Value	AK	A0	*p*-Value
TEWL						
Forearm	−0.302 (−0.489, 0.0149)	0.0058 (−0.12, 0.081)	0.018	−0.529 (−0.601, −0.428)	0.148 (−0.571, 0.578)	<0.001
Leg	−0.0976 (−0.321, 0.244)	0.0143 (−0.248, 0.116)	0.854	−0.288 (−0.333, −0.176)	0.507 (0.0656, 1.36)	<0.001
Forehead	−0.220 (−0.375, −0.0448)	−0.0128 (−0.196, 0.152)	0.036	−0.457 (−0.612, −0.243)	0.150 (−0.0571, 0.636)	<0.001
Hydration						
Forearm	−0.0196 (−0.0784, 0.0959)	−0.184 (−0.256, −0.0132)	0.034	0.452 (0.300, 0.560)	−0.0810 (−0.282, 0.119)	0.001
Leg	0.143 (−0.134, 0.212)	−0.0270 (−0.246, 0.0896)	0.320	0.250 (0.213, 0.522)	0.0470 (−0.0894, 0.220)	0.034
Forehead	0.128 (−0.0447, 0.373)	−0.127 (−0.325, 0.0356)	0.012	0.244 (0.146, 0.537)	0.0450 (−0.297, 0.466)	0.086
Erythema						
Forearm	−0.0556 (−0.186, −0.0170)	−0.0745 (−0.111, 0.0119)	0.685	−0.133 (−0.185, −0.0692)	−0.0404 (−0.191, 0.129)	0.221

Q1—first quartile; Q3—third quartile. HK—healthy skin with kefir intake; H0—healthy skin without kefir intake; AK—atopic skin with kefir intake; A0—atopic skin without kefir intake. TEWL—transepidermal water loss. Groups compared by Mann-Whitney U test, with *p* < 0.05 for statistical significance.

**Table 4 foods-10-02794-t004:** Variation of skin parameters, after lesion induction with SLS, for the healthy group (median (Q1, Q3)).

Deviation of Skin Parameters at Forearm	Healthy Group (*n* = 33)		
	HK	H0	*p*-Value
TEWL SLS	−0.2931 (−0.510, −0.180)	0.0878 (−0.0924, 0.243)	<0.001
Hydration SLS	0.0000 (−0.133, 0.106)	0.0065 (−0.0889, 0.138)	0.347
Erythema SLS	−0.0287 (−0.0371, 0.0644)	0.0144 (−0.0775, 0.135)	0.825

Q1—first quartile; Q3—third quartile. Deviation (variable) = [(variable at t8) – (variable at t0)]/(variable at t0). HK—healthy skin with kefir intake; H0—healthy skin without kefir intake; TEWL—transepidermal water loss. SLS—sodium lauryl sulphate. Groups compared by Mann-Whitney U test, with *p* < 0.05 for statistical significance.

**Table 5 foods-10-02794-t005:** Comparison of SCORAD Index variation, between kefir intake and control, for atopic group, after intervention (median (Q1, Q3)).

Deviation of SCORAD Index	Atopic Group (*n* = 19)		
	AK	A0	*p*-Value
SCORAD	−0.626 (−0.758, −0.491)	0.0402 (−0.0293, 0.273)	<0.001

Q1—first quartile; Q3—third quartile. SCORAD—SCORing of Atopic Dermatitis. AK—atopic skin with kefir intake; A0—atopic skin without kefir intake. Groups compared by Mann-Whitney U test, with *p* < 0.05 for statistical significance.

**Table 6 foods-10-02794-t006:** Multiple linear regression for effect of kefir status on skin parameters (standardized regression coefficient ß (*p*-value), *n* = 52).

Deviation of Skin Parameters	β for Kefir Intake (*p*-Value)		
	Model 1	Model 2	Model 3
TEWL			
Forearm	−0.596 (<0.001)	−0.597 (<0.001)	−0.625 (<0.001)
Leg	−0.304 (0.029)	−0.323 (0.018)	−0.332 (0.020)
Forehead	−0.501 (<0.001)	−0.502 (<0.001)	−0.524 (<0.001)
Hydration			
Forearm	0.481 (<0.001)	0.458 (<0.001)	0.539 (<0.001) a
Leg	0.294 (0.034)	0.267 (0.042)	0.347 (0.006) a
Forehead	0.362 (0.008)	0.346 (0.011)	0.358 (0.012)
SCORAD Index (**)	−0.910 (<0.001)	n.a.	−0.866 (<0.001)

β—standardized regression coefficient (reference category: without kefir intake), *p* < 0.05 for statistical significance. Model 1—kefir Status; Model 2—kefir status, skin status; Model 3—kefir status, skin status, gender; water intake. (**) Variable skin status was excluded from the models. n.a.—not applicable. a—gender and water contribution showed *p* < 0.05. TEWL—transepidermal water loss. SCORAD—SCORing of Atopic Dermatitis.

## Data Availability

Data are available from the corresponding author upon reasonable request.

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
