# Peer review of "Homemade Kefir Consumption Improves Skin Condition—A Study Conducted in Healthy and Atopic Volunteers"

_foods, 2021, doi:10.3390/foods10112794_

Round 1

Reviewer 1 Report

Dear all,

below are my comments and suggestions

Manuscript ID: foods-1449901

The research entitled " Homemade kefir consumption improves skin condition – a study conducted in healthy and atopic volunteers" is interesting and useful research in the direction of providing information on the intake effects of homemade kefir in the skin of healthy and atopic volunteers and the improvement in the degree of severity of atopic dermatitis.

However, some changes need to be made in the manuscript itself

Introduction: The introduction can be improved and include more relevant references regarding kefir properties and applications.

The number of bibliographic sources is adequate, 50% of the total bibliographic sources are from the last 5 years. All cited literature is relevant to the field addressed in the paper.

There are some grammatical errors and instances of badly worded/constructed sentences throughout the manuscript. Please refine the language carefully.

Author Response

Manuscript ID: foods-1449901

Title: Homemade kefir consumption improves skin condition – a study conducted in healthy and atopic volunteers

Reply to reviewer 1

Authors answer

We appreciate all your comments that have been taken into account.

The Introduction section includes now the antioxidant capacity of kefir as well as topical application on wound healing. The appropriate references have been added.

In addition, a review of the English language was carried out.

Reviewer 2 Report

Previous studies showed that kefir is a rich probiotic, having protective effect, thanks to its antioxidant, anti-inflammatory, and immunomodulatory capacity. The UHPLC screening of kefir water revealed that the major twenty metabolites found were naturally occurring flavonoids and phenolic derivative. These listed flavonoids and phenolic acid derivatives have been studied previously for their high antioxidant, anti-inflammation, and anti-cancer properties Previous studies have also demonstrated the ability of Lactobacillus species in metabolising flavonoid and phenolic compounds as their end products in fermentation. However, this ability is based on species- or strain specific. The enzymatic hydrolysis of lactic acid bacteria fermentation increases both the flavonoid and phenolic compound production; thus, increases the antioxidant activities in the kefir. In numerous experimental models, natural antioxidants are shown  to induce hormetic dose responses that are not only common but display endpoints of biomedical and clinical relevance. These hormetic responses are mediated via the activation of nuclear factor erythroid- derived 2 (Nrf2) antioxidant response elements (AREs) and, as such, are characteristically biphasic, well integrated, concentration/dose dependent, and specific with regard to the targeted cell type and the temporal profile of response. In experimental disease models, the polyphenol-induced hormetic activation of Nrf2 was shown to effectively reduce the occurrence and severity of a wide range of human-related pathologies, including Parkinson's disease, Alzheimer's disease, stroke, age-related ocular damage, chemically induced brain damage, and renal nephropathy, amongst others, while also enhancing stem cell proliferation. Thus, Interplay and coordination of redox interactions with endogenous and exogenous antioxidant defence systems  is an emerging area of reserach interest in anticancer and antidegenerative therapeutics.

This is an interesting paper.  The study is well-conceived and well-executed. This reviewer is satisfied with the significance of this study, the care in which the study was performed, and the implications of the results for human health.  However, although the results presented are convincing, the work raises some concerns which will need to be addressed. The questions posed are of extremely high interest, but the paper does not give adequate definitive information.

Minor concerns:

  1. Given the relationship between polyphenol compounds, redox status and the vitagene network and its possible biological relevance in neuroprotection, Authors while interpetrating results should discuss appropriately this aspect and make proper connection with emerging principles of hormesis. Indeed, preconditioning signal leading to cellular protection through Hormesis is an important redox dependent aging-associated to free radicals species accumulation, inflammatory responses involved in neurodegenerative/ neuroprotective mechanisms. This aspect should be highlighted in the discussion and references properly added.

Author Response

Manuscript ID: foods-1449901

Title: Homemade kefir consumption improves skin condition – a study conducted in healthy and atopic volunteers

Reply to reviewer 2

Authors answer

We appreciate all your comments that have been taken into account.

The Discussion section includes now the concept of hormesis, as well as the antioxidant capacity of the probiotic LAB, present in the milk-kefir drink.

The appropriate references have been added.
